# New Insights into the Interaction between Graphene Oxide and Beta-Blockers

**DOI:** 10.3390/nano9101429

**Published:** 2019-10-09

**Authors:** Yuehua Deng, Yani Li, Wenjie Nie, Xiang Gao, Shentan Liu, Xiaochun Tan, Mingming Chen, Dongzhuang Hou

**Affiliations:** 1College of Geology and Environment, Xi’an University of Science and Technology, Xi’an 710054, China; ynli2019@163.com (Y.L.); nwj@xust.edu.cn (W.N.); gaoxiang@iccas.ac.cn (X.G.); liushentan@seu.edu.cn (S.L.); txc9150@163.com (X.T.); 13720599719@163.com (M.C.); houdongzhuang0207@163.com (D.H.); 2Shaanxi Provincial Key Laboratory of Geological Support for Coal Green Exploitation, Xi’an 710054, China

**Keywords:** magnetic graphene oxide, propranolol, Fe_3_O_4_, adsorption

## Abstract

As a nano-adsorbent, magnetic graphene oxide (GO/Fe_3_O_4_) was synthesized to potentially adsorb propranolol (PRO) from water. The synthetic material was characterized by SEM, TEM, VSM, FTIR, XRD, zeta potential, and XPS. The environmental factors, such as pH, humic acid concentration, PRO concentration, and contact time, were investigated regarding their effect on the adsorption process. The kinetics data fitted the pseudo first-order and second-order kinetics equations. The Langmuir equation, the Freundlich equation, and the Sips equation were used to analyze the adsorption isotherms. Electrostatic attraction, hydrogen bonding, and the π–π interaction all contributed to the adsorption process of PRO onto GO/Fe_3_O_4_. The discovery of this study emphasized the feasibility of GO/Fe_3_O_4_ removal of PRO and expanded the scope of the application of GO.

## 1. Introduction

Graphene, as a member of the group of carbon materials, has attracted enormous attention from researchers since 2004 due to its excellent mechanical and physicochemical properties [1]. As its derivative, graphene oxide (GO) exhibits superior properties. Both GO and graphene have two-dimensional, sheet-like structures which are connected by carbon atoms through the hybridization of sp^2^ and σ bonds to the surrounding carbon atoms to form a hexagonal honeycomb lattice [2,3,4]. Compared with graphene, GO has a large number of oxygen-containing functional groups at the edges and inside of the sheet, including hydroxyl groups, carbonyl groups, carboxyl groups, and epoxy groups, which allows GO to be operatively bonded to organic pollutants [5,6,7]. Therefore, GO is often used as an adsorbent in wastewater treatment.

The environmental residue of pharmaceutical and personal care products (PPCPs) has received increasing attention worldwide [8,9]. As a new type of pollutant, propranolol (PRO) is a beta-blocker for the treatment of cardiovascular disease [10]. After entering the water body as a parent compound, it cannot be completely removed by the traditional sewage treatment process [11,12]. Toxicity tests indicated that PRO remaining in water can cause damage to aquatic organisms and even harm human health [13,14,15,16]. Kyzas et al. studied the removal of PRO and atenolol from water by GO adsorption [17]. The results showed that the oxygen-containing functional groups on the surface of GO can effectively remove atenolol and PRO from water via electrostatic interaction and hydrogen bonding. However, the presence of these groups on the surface of GO has a double-sided effect. It is easily bonded to contaminants, but it is difficult for GO to be separated from water bodies due to the hydrophilicity of these groups [18,19]. After the adsorption process completes, it must be separated from the aqueous solution via filtration or centrifugation, which is complicated, time-consuming, and costly.

Magnetic separation is an effective way to solve this problem. Magnetic nanoparticles have received extensive attention due to their unique magnetic properties. A specific external magnetic field is applied, the magnetic material can be separated according to a specific path. The magnet disappears and the material can be easily dispersed into the solution again when the applied magnetic field is removed [20,21]. Based on this advantage, combining Fe_3_O_4_ with GO not only preserves the high adsorption capacity of GO, but also allows it to be quickly separated from water. It is worth mentioning that the loading of Fe_3_O_4_ particles makes the GO sheets less prone to aggregation, which is due to the presence of van der Waals forces between the GO layers [22,23]. Miao et al. showed that GO/Fe_3_O_4_ has a high adsorption capacity for adsorbing tetracyclines [24]. He et al. formed GO-Fe_3_O_4_ hybrid by covalent bonding to adsorb methylene blue and neutral red cationic dyes with a high adsorption capacity [25]. Huang et al. studied the adsorption of tetracycline, Cd, and arsenate (As(V)) on graphene oxide when they coexist, and the excellent property of GO/Fe_3_O_4_ was mainly attributed to its high dispersibility, thin nanosheets and various functional groups [26]. The above researches indicated the possibility of using magnetic GO in the field of wastewater treatment in the future. As far as is known, however, there has been no literature on the removal of PRO from water by GO/Fe_3_O_4_.

In this study, GO was prepared using the modified Hummers method and GO/Fe_3_O_4_ was further synthesized in a one-step reaction. PRO was chosen as the target contaminant. The factors affecting PRO adsorption were systematically investigated, and the adsorption kinetics and adsorption isotherms for PRO were studied in detail. The objectives of this study were as follows: (1) prepare GO/Fe_3_O_4_ magnetic composites, (2) study the adsorption behavior of PRO onto GO/Fe_3_O_4_, and (3) discuss the possible adsorption mechanisms of this process.

## 2. Materials and Methods 

### 2.1. Materials

Propranolol hydrochloride used in the experiment was purchased from Aladdin Reagent Co., Ltd. (Shanghai, China). Graphite powder was supplied by Guangdong Xilong Chem. Co. Ltd. (Shantou, China). The formic acid and acetonitrile were of HPLC grade. All other reagents, including humic acid (HA) and ferrous sulfate heptahydrate (FeSO_4_·7H_2_O), were analytical grade and used without any further purification. All solutions were prepared using distilled water. 

### 2.2. Preparation of GO/Fe_3_O_4_

The preparation method of the material referred to our previous research [27].

Preparation of GO: 1 g each of potassium sulfate, phosphorus pentoxide, and graphite powder were vigorously stirred in 10 mL of concentrated sulfuric acid and kept at 80 °C for 5 h. It was then washed with distilled water to a neutral pH and dried at 60 °C in a vacuum oven. Potassium permanganate (4 g) and the above product were dissolved into concentrated sulfuric acid (40 mL) placed in an ice bath. After keeping at 35 °C for 2 h, distilled water (100 mL) was slowly added to the mixture, and the mixture was heated to 98 °C for 15 min. Hydrogen peroxide was added to neutralize excess potassium permanganate until the solution was a golden yellow. After settling, the supernatant was decanted, and the precipitant was washed with dilute hydrochloric acid and distilled water to a neutral pH. Finally, the mixture was dried at 60 °C and passed through a 200-mesh screen.

Preparation of GO/Fe_3_O_4_: GO (0.5 g) was dispersed into deionized water and sonicated for 30 min. Then, the mixture was kept in a N_2_ atmosphere for 15 min. Ferrous sulfate heptahydrate (2 g) was added to the above solution under N_2_ bubbling at 90 °C. The mixture was labeled as solution A. Sodium hydroxide (1.8 g) and sodium nitrate (0.9 g) were dissolved in distilled water (40 mL), and the solution was marked as solution B. Solution B was added dropwise to solution A under N_2_ bubbling. Then, the final mixture was kept at 90 °C for 4 h. After being cooled to room temperature, it was washed to a neutral pH, dried at 60 °C, and passed through a 100-mesh screen.

### 2.3. Characterization of GO/Fe_3_O_4_

The surface topography and structure of the prepared materials were observed using SEM (JEOL, Tokyo, Japan) and TEM (FEI-JSM 6320F, JEOL, Tokyo, Japan). The magnetic property of the material was measured using VSM (Lake Shore, USA) at room temperature. The key functional groups of the adsorbent were analyzed using FTIR (Thermo Electron Nicolet-360, USA). The XRD patterns of material were conducted to determine the crystal structure with X-ray diffractometer (ARL Co., Switzerland) at the range of 5° to 90°. The surface charge of the adsorbent was measured at various pH values using the zeta potential (Malvern Instrument Co., U.K.). The surface elements of the adsorbent were analyzed using XPS.

### 2.4. Batch Experiments

All experiments were conducted to evaluate the adsorption performance of GO/Fe_3_O_4_ at 30 °C. The amount of PRO stock solution (1 g/L) was added into the centrifuge tube with 0.01 g adsorbent. The effects of adsorption time, the concentration of PRO, pH, and the concentration of humic acid (HA) were studied in turn. The pH of the reaction system was adjusted using 0.1 mol/L NaOH and 0.1 mol/L HCl solutions. The mixtures were shaken for a given time. The solution passed through a 0.22 μm filter with a syringe after equilibrium. The initial and final concentrations of PRO were measured by HPLC at a wavelength of 290 nm. The adsorption capacity was determined using Equation (1):(1)$$q_{e}=\frac{\left(C_{0}-C_{e}\right)V}{m}$$
where *C_0_* (mg/L) and *Ce* (mg/L) are the initial and equilibrium concentrations of PRO. *q_e_* (mg/g) is the adsorption capacity of GO/Fe_3_O_4_. *V* (L) is the volume of the solution and *m* (g) represents the mass of GO/Fe_3_O_4_. 

## 3. Results and Discussion

### 3.1. Characterization of GO/Fe_3_O_4_

In order to study the morphological and microstructural details of GO/Fe_3_O_4_, SEM images of GO and GO/Fe_3_O_4_ were conducted and presented in Figure 1a,b. A TEM image of GO/Fe_3_O_4_ was shown in Figure 1c. 

As shown in Figure 1a, GO exhibited a sheet-like structure with a smooth surface and edges, which was consistent with that described in previous reports. Partial aggregation can also be observed due to the existence of oxygen-containing functional groups on the GO sheet. Figure 1b showed the successful formation of the compound. It can be seen that the GO was the basal plane of prepared material and Fe_3_O_4_ particles were uniformly distributed and attached on the surface of GO. The formation of the GO/Fe_3_O_4_ did not cause damage to the structure of GO and GO maintained its original morphology, which suggested that GO/Fe_3_O_4_ still kept the unique properties of GO while being quickly recycled due to the presence of Fe_3_O_4_ particles. Some wrinkles were observed on the GO surface and provided a larger surface area for the successful loading of Fe_3_O_4_ particles, preventing from the aggregation of GO. The surface coverage of the prepared material was further identified using the TEM image (Figure 1c). It was also found that Fe_3_O_4_ particles (about 30 nm) were loaded on the transparent and slightly aggregated GO films.

The magnetic property of GO/Fe_3_O_4_ was researched by measuring the field-dependent magnetization curve at room temperature. As shown in Figure 2, the saturation magnetization of Fe_3_O_4_/GO was as high as 53.57 emu/g, manifesting the high magnetic performance of the prepared material. The magnetization of Fe_3_O_4_/GO increased with the increase of the applied magnetic field strength and increased to a saturation value. The hysteresis loop was a smooth S-shape and coincided well without remanence magnetization, which indicated that the Fe_3_O_4_/GO was superparamagnetic and was easy to be collected from the adsorption process.

FTIR spectrum was used to identify the characteristic functional groups of GO/Fe_3_O_4_. Figure 3 presented the FTIR of GO, GO/Fe_3_O_4_, and GO/Fe_3_O_4_ after adsorption. In the infrared spectrum of GO, the broad peak at 3424 cm^−1^ represented the O–H stretching vibration, which formed a hydrogen bond with an oxygen atom of the adsorbed water molecule. The stretching vibration of the C=O bond in the carboxyl group was identified via the peak at 1721cm^−1^. The peak at 1622 cm^−1^ was associated with the stretching vibration of C=C of the aromatic ring. The peaks at the 1381 cm^−1^, 1220 cm^−1^, and 1051 cm^−1^ were attributed to the stretching vibrations of C–H and C–O bonds. The functional groups on the surface and edges of GO improved its hydrophilicity. Compared to the GO spectrum, some peaks had slightly changed in the spectrum of GO/Fe_3_O_4_ due to the attachment of Fe_3_O_4_ particles. The weakened peak observed at 3424 cm^−1^ may be attributed to an increase of Fe_3_O_4_ and a decrease of GO in the prepared material. Another strong peak at 583 cm^−1^ may be due to the fact that an overlap occurred between a peak of GO at 582 cm^−1^ and the characteristic peak of Fe_3_O_4_ at 583 cm^−1^, which demonstrated that Fe_3_O_4_ particles were successfully loaded onto the GO. A significant vibration band formed at 1400 cm^−1^, which was attributed to the formation of either a monodentate complex or a bidentate complex between the carboxyl group and Fe on the surface of the Fe_3_O_4_ particles [28]. After PRO adsorption onto GO/Fe_3_O_4_, new peaks appeared at 1105 cm^−1^ and 1069 cm^−1^, signifying the C–O–C stretching of the aryl alkyl ether in PRO. A weak peak at 2959 cm^−1^ was connected with N-H stretching in secondary amines of PRO. A significant peak at 1264 cm^−1^ represented the formation of hydrogen band between the oxygen atom in the epoxy group and the PRO molecule.

Figure 4 displayed the XRD patterns of GO, GO/Fe_3_O_4_, and GO/Fe_3_O_4_ after adsorption. For the GO pattern, a sharp diffraction peak at 10.8° was observed, manifesting the presence of residual stacked layers of GO [29]. The diffraction peaks at 2θ values of 35.6°, 43.4°, 53.7°, and 62.8° were consistent with the characteristic peaks of Fe_3_O_4_ in GO/Fe_3_O_4_, which revealed the successful attachment of Fe_3_O_4_ particles. However, the sharp peak did not appear in the composite, indicating that there was a thin layer structure caused by the Fe_3_O_4_ particles. It can be noticed there were no significant changes after PRO adsorption, which was attributed to the fact that PRO adsorption caused a reduction in the relative content of Fe_3_O_4_ particles.

Zeta potential of GO, Fe_3_O_4_, and GO/Fe_3_O_4_ were exhibited in Figure 5. The zeta potential of GO was negative at the test pH range, illustrating that the surface of GO was charged negatively. The zero-potential point of Fe_3_O_4_ was about pH = 6. When pH < 6, the surface of the Fe_3_O_4_ was positively charged. When pH > 6, negative charges dominated on the surface of Fe_3_O_4_. It can be observed that the values of GO/Fe_3_O_4_ were between GO and Fe_3_O_4_, indicating that the surface charge of GO was changed due to the loading of Fe_3_O_4_.

The XPS spectra of GO/Fe_3_O_4_ before and after the adsorption of PRO with binding energies ranging from 0 to 1400 eV was acquired for the identification of the surface elements and performance of a quantitative analysis. It was obvious in Figure 6a that the peaks of Fe 2p, O 1s, and C 1s were found in the full scan spectrums before and after adsorption of PRO, which suggested the existence of Fe, O, and C in the GO/Fe_3_O_4_. For the high-resolution Fe 2p of GO/Fe_3_O_4_ (Figure 6b), the binding energies of 711.38 and 725.38 eV corresponded to Fe^2+^ and Fe^3+^, which indicated the successful synthesis of Fe_3_O_4_ nanoparticles and loading onto the surface of GO. The finding was consistent with the above characterization results. From Figure 6c,d, the C=O, C–O, and C=C/C–C characteristic bonds were present at around 284.5, 285, and 286.5 eV, respectively. It can be observed that the peak intensity of C=C/C–C after adsorption became stronger than that of C=C/C–C before adsorption, which was attributed to the introduction of PRO. The elemental content was given in Table 1. It can be seen that the carbon content after adsorption was significantly increased compared with before adsorption, which also indicated that PRO was successfully adsorbed onto GO/Fe_3_O_4_. The elemental carbon contained in PRO led to an increase in the total carbon content. The successful adsorption of PRO was also illustrated by nitrogen. The content of nitrogen increased from 0.47 to 0.87, which was caused by the amino group in PRO. At the same time, it can be seen that the content of Fe was reduced, and it was speculated that the decrease of Fe was caused by the leaching of Fe during the reaction, which was discussed in detail below. The increase in the content of other elements, especially carbon, led to a relative decrease in the proportion of oxygen.

### 3.2. Adsorption Kinetics

The study of adsorption kinetics for PRO onto GO/Fe_3_O_4_ was important for understanding the rate and mechanism of the adsorption process. The effect of contact time on adsorption was depicted in Figure 7. It can be seen that the reaction was very initially rapid and the adsorption capacity of PRO increased dramatically with time increasing. There were large surface area and a number of active sites on the surface of GO/Fe_3_O_4_, which was good for the adsorption process. The high concentration of PRO in an aqueous solution was also attributed to the rapid adsorption. After 60 min, the adsorption rate gradually decreased. The reason for this trend was that the adsorption sites on the surface of the material were gradually occupied by PRO and the availability of the active sites was reduced, resulting in a slow adsorption rate. Moreover, the adsorbed PRO on the surface of GO/Fe_3_O_4_ may have also hindered the adsorption of PRO in the aqueous solution due to steric hindrance. With time increasing, the adsorption of PRO onto GO/Fe_3_O_4_ was not significantly affected by the contact time and the adsorption tended to be in a stable state. 

Experimental data of the adsorption of PRO onto GO/Fe_3_O_4_ was fitted using a pseudo-first-order kinetics model (Equation (2)) and a pseudo-second-order kinetics model (Equation (3)):(2)$$ln\left(q_{e}-q_{t}\right)=lnq_{e}-K_{1}t$$
(3)$$\frac{t}{q_{t}}=\frac{1}{K_{2}q_{e}^{2}}+\frac{t}{q_{e}}$$
where *q_e_* and *q_t_* are the amount of PRO adsorbed onto the GO/Fe_3_O_4_ (mg/g) at equilibrium and at time t (min), respectively; *K_1_* is the rate constant of pseudo-first-order adsorption (1/min); and *K_2_* is the constant of pseudo-second-order rate (g/(mg·min)). The kinetics curves were shown in Figure 8. The theoretical data and the fitted parameters were shown in Table 2. 

The correlation coefficient R^2^ for the pseudo-first-order kinetics model was 0.338 and the experimental data (30.34 mg/g) was quite different from the theoretical calculation data (6.58 mg/g). It can be judged that the model did not fit the experimental data well. The same result was also obtained from Figure 8a. Compared with the pseudo-first-order kinetics model, the correlation coefficient R^2^ for the pseudo-second-order kinetics model was higher (R^2^=0.999) and the experimental data was consistent with calculated values. It was observed directly from Figure 8b that the adsorption of PRO onto GO/Fe_3_O_4_ obeyed the pseudo-second-order kinetics model. The finding suggested that the control of the strong chemical reaction or surface complexion determined the adsorption rate.

### 3.3. Adsorption Isotherm

The adsorption isotherm experiment was conducted to get a better understanding of the adsorption mechanism of PRO onto GO/Fe_3_O_4_. The relationship curve for the PRO equilibrium concentration and adsorption capacity was presented in Figure 9. It can be seen that the adsorption capacity increased linearly with the PRO initial concentration increasing at low concentrations, then showed a smaller increase when the concentration was further increased, and finally maintained a stable value. The results indicated that the PRO concentration was of great importance to the adsorption process. In the beginning, GO/Fe_3_O_4_ showed a strong affinity to PRO due to the sufficient adsorption sites on the surface of adsorbents. The PRO transportation from the solvent to GO/Fe_3_O_4_ surface was promoted by the increase of the PRO concentration. During the process of adsorption, the decrease of active sites on the surface of adsorbents led to the adsorption capacity being constant. 

Experimental data were fitted by Langmuir, Freundlich, and Sips isotherm models, respectively. 

The Langmuir model usually describes a monolayer adsorption, where the adsorption process occurs at homogeneous sites and the adsorbed molecules do not affect each other. The linear form of the Langmuir equation s expressed in Equation (4):(4)$$\frac{1}{q_{e}}=\frac{1}{q_{m}}+\frac{1}{K_{L}q_{m}C_{e}}$$
where *q_m_* (mg/g) is the maximum amount of the adsorbed PRO per unit mass of GO/Fe_3_O_4_; *Ce* (mg/L) is the equilibrium concentration of PRO onto the adsorbent; *q_e_* (mg/g) is the amount of the adsorbed PRO at equilibrium; and *K_L_* is the Langmuir adsorption constant, which relates to the affinity and adsorption energy of the bonding sites.

The Freundlich model is an empirical relationship model that describes the adsorption process of heterogeneous surface. The Freundlich model is expressed in Equation (5): (5)$$\log q_{e}=\frac{1}{n}\log C_{e}+\log K_{F}$$
where *q_e_* (mg/g) is the amount of the adsorbed PRO at equilibrium; *Ce* (mg/L) is the equilibrium concentration of PRO onto the adsorbent; *K_F_* is Freundlich constant, which relates to the adsorption capacity and the adsorption strength; and 1/*n* is the heterogeneity factor, where the value of 1/*n* is between 0 and 1, and characterizes the effect of the concentration on the amount of adsorption. 

The Sips isotherm model was developed on the basis of the Langmuir and Freundlich models. The introduction of the third parameter gives it a wider range of applications. The Sips model is presented in Equation (6):(6)$$q_{e}=\frac{q_{m}{\left(bC_{e}\right)}^{1/n}}{1+{\left(bC_{e}\right)}^{1/n}}$$
where *q_e_* (mg/g) is the amount of the adsorbed PRO at equilibrium; *q_m_* (mg/g) is the maximum amount of the adsorbed PRO per unit mass of GO/Fe_3_O_4_; *Ce* (mg/L) is the equilibrium concentration of PRO onto the adsorbent; *1/n* is the heterogeneity factor, where the value for 1/*n* getting closer to 1 indicates a relatively homogenous adsorbent surface; and *b* (L/mg) is the median association constant. 

Experimental data were fitted using the three isotherm models described and the obtained curves were shown in Figure 9. The fitting results were shown in Table 3. It was clear that the Langmuir adsorption isotherms had the lowest correlation coefficients (0.857) among the three models, which indicated that the adsorption process was not a single monolayer adsorption onto GO/Fe_3_O_4_ and the distribution of sites on the adsorbent was not homogeneous. The findings for this experiment were consistent with the fitting parameter values of the Freundlich model. Compared to the Langmuir model, the Freundlich adsorption isotherm had a higher R^2^ value (0.961). The value of *n* was between 1 and 10, indicating that the occurrence of the adsorption process was favorable. When the PRO concentration was low, the experimental data were better fitted with the theoretical data calculated by the Freundlich model. However, the Freundlich adsorption isotherm deviated from the experimental data as the PRO concentration increased. The Sips adsorption isotherm was more appropriate for the adsorption of PRO onto GO/Fe_3_O_4_ and the R^2^ value (0.983) was the highest amongst the three involved models. For the Sips isotherm, 1/*n* is less than 1, indicating a heterogeneous adsorbent.

### 3.4. Effect of pH 

The pH of the solution determines not only the surface charge of the adsorbent but also the existing form of the adsorbate. Therefore, the effect of pH on the adsorption process was investigated and the results are given in Figure 10a. PRO was positively charged when the pH of the solution was less than 9.5. At a pH below 5, the adsorption capacity of PRO onto GO/Fe_3_O_4_ increased rapidly, which was ascribed to an increasing of H^+^ concentration with pH increasing. Under a strong acid condition, a high concentration of H^+^ occupied the adsorption sites on the surface of the adsorbent. The adsorption of PRO was suppressed due to the competitive adsorption of H^+^. However, the adsorption inhibition of PRO onto GO/Fe_3_O_4_ by H^+^ was weakened as the pH increased. It can be seen from Figure 5 that GO/Fe_3_O_4_ was positively charged and there was an electrostatic repulsion between the adsorbent and PRO, which may have been responsible for the lower adsorption capacity of PRO under acidic conditions. When the pH was further increased, a slowly increasing trend of adsorption capacity was presented. The hindrance of H^+^ was weaker. Furthermore, GO/Fe_3_O_4_ was negatively charged when pH > 6. Thus, there was an electrostatic attraction between PRO and the adsorbent, which contributed to the adsorption of PRO. As shown in Figure 10b, the pH values after the reaction were higher than those before the reaction when the pH was below 7. The decrease of the H^+^ concentration suggested that H^+^ participated in the adsorption process of PRO. When the pH was increased from 8 to 9.5, the adsorption capacity increased rapidly and reached a maximum adsorption capacity. Adsorption competition of H^+^ was weak, and other forces, like hydrogen bonding and π–π interactions between PRO and the adsorbent, may have dominated the adsorption process. When pH > pKa, the amino group of PRO was deprotonated. PRO existed as a neutral molecule. The weakening of the electrostatic attraction led to a slow decrease of the adsorption capacity. The adsorption of PRO onto GO/Fe_3_O_4_ had a similar adsorption tendency to that on GO (Appendix A in the Appendix A), indicating that the adsorption mechanism of GO/Fe_3_O_4_ mainly depended on GO.

### 3.5. Material Stability

To determine the stability of the material, the amount of Fe in the solution was determined after the adsorption equilibrium was reached. Figure 11 showed the amount of Fe leaching at different pH levels. When the pH was between 4 and 9, the Fe element was hardly leached, but was easily leached under strong acid or alkali conditions. The results suggested that GO/Fe_3_O_4_ can be used in a wide range of neutral water.

### 3.6. Effect of HA

HA, widely distributed in water, can interact with organic matter. HA has an important influence on the migration behavior of PRO in the environment. The effect of HA on the adsorption of PRO onto GO/Fe_3_O_4_ under different HA concentrations is shown in Figure 12. The presence of HA had a little effect on the adsorption of PRO onto GO/Fe_3_O_4_. The adsorption capacity dropped slightly and then remained constant. HA surface was negatively charged as the pH of the solution was above 2 [30]. The pH condition for this experiment was 7.5. Thus, both HA and GO/Fe_3_O_4_ were negatively charged. The electrostatic repulsion between them led to HA tending to be stable in the aqueous solution rather than being adsorbed onto the GO/Fe_3_O_4_ surface. The electrostatic attraction between the cationic PRO and the negatively charged HA caused the PRO adsorbed onto GO/Fe_3_O_4_ to desorb from its surface into the aqueous solution, which hindered the adsorption of PRO onto GO/Fe_3_O_4_. The initial concentration of PRO in this experiment was small, thus the solubilization was not significant.

### 3.7. Comparison of GO/Fe_3_O_4_ with other Adsorbents

To further evaluate the performance of the adsorbent, the adsorption capacity of GO/Fe_3_O_4_ was compared to other adsorbents. The adsorption capacity of PRO onto GO was 67 mg/g [17], and the adsorption capacity of GO/Fe_3_O_4_ was lower relatively due to the introduction of Fe_3_O_4_ nanoparticles. However, the loading of Fe_3_O_4_ greatly reduced the manufacturing cost of the adsorbent and facilitated the separation of GO from water. In the previous work, the adsorption of PRO onto acidified attapulgite (48.05 mg/g), chitosan-modified attapulgite (26.38 mg/g), and coupling-agent-modified attapulgite (24.56 mg/g) were also studied [16]. It is obvious that the adsorption capacity of PRO onto GO/Fe_3_O_4_ was higher than the above adsorbents. In summary, GO/Fe_3_O_4_ could be used as a promising adsorbent to remove PRO from water.

## 4. Conclusions

A nanomagnetic material GO/Fe_3_O_4_ was prepared facilely and investigated regarding its ability to remove contaminant PRO from water via adsorption. All the characterization results indicated that the prepared adsorbent was successfully synthesized and easily recovered from the water by an external magnetic field. The kinetics experimental data were fitted well with the pseudo-second-order kinetics, showing that the adsorption process was controlled by chemical reactions or surface complexation. The high correlation coefficient of the Sips model showed multilayer adsorption and the heterogeneous adsorbent surface. Electrostatic attraction, hydrogen bonding, and π–π interactions all contributed to the adsorption process of PRO onto GO/Fe_3_O_4_. The adsorption process was pH-dependent and slightly inhibited by HA. All experimental results revealed that GO/Fe_3_O_4_ can be used as an effective adsorbent for the removal of PRO from water.

## Figures and Tables

**Figure 1 nanomaterials-09-01429-f001:**
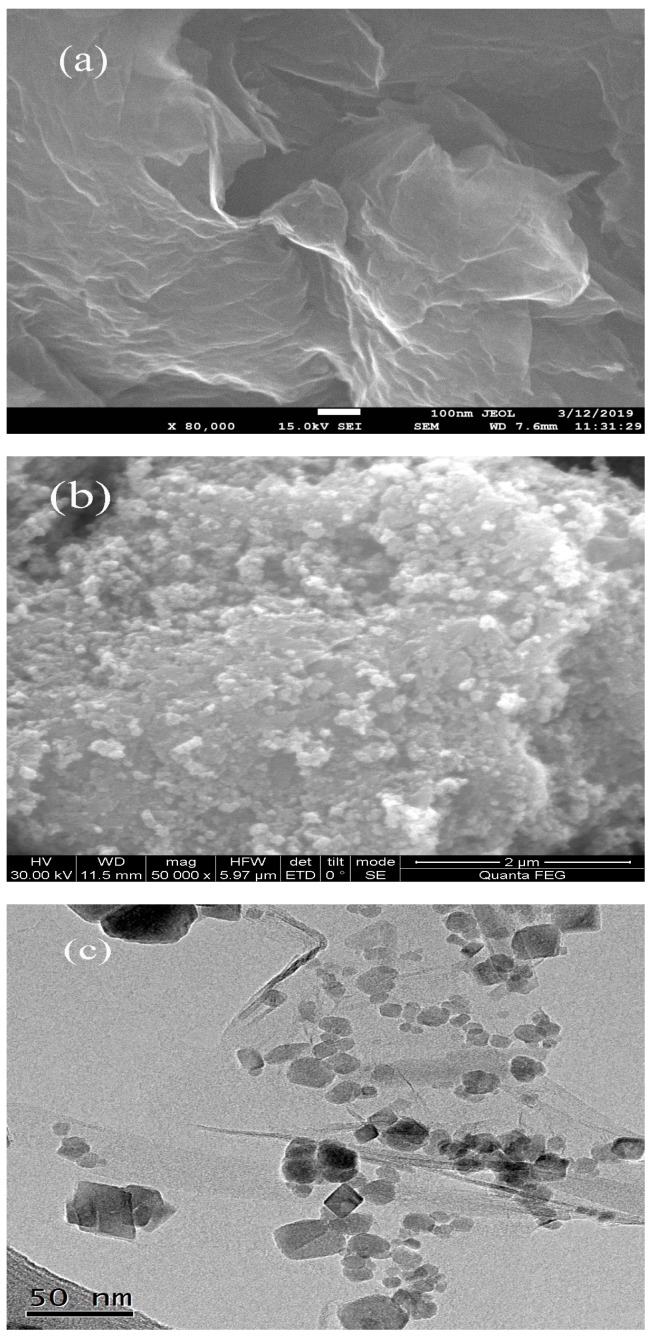
SEM images of GO (**a**) and GO/Fe_3_O_4_ (**b**), and TEM image of GO/Fe_3_O_4_ (**c**).

**Figure 2 nanomaterials-09-01429-f002:**
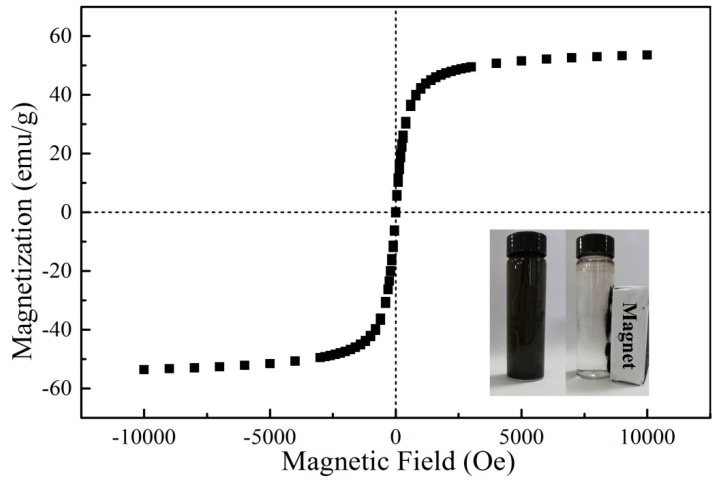
Magnetization curve of GO/Fe_3_O_4_ at room temperature.

**Figure 3 nanomaterials-09-01429-f003:**
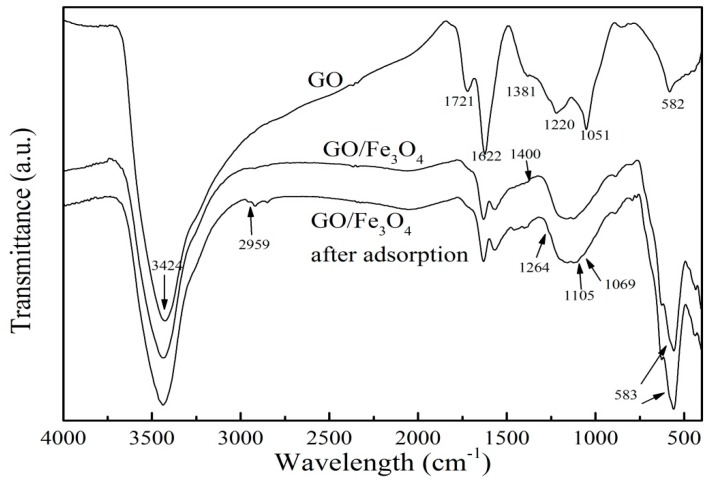
FTIR spectrum of GO, GO/Fe_3_O_4_, and GO/Fe_3_O_4_ after adsorption.

**Figure 4 nanomaterials-09-01429-f004:**
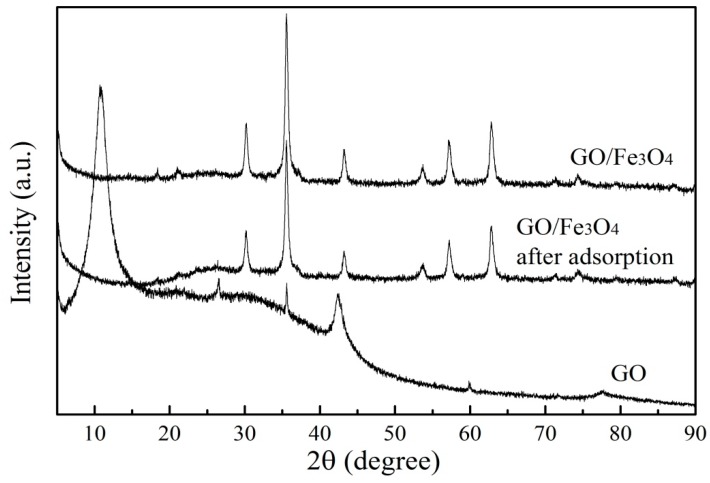
XRD of GO, GO/Fe_3_O_4_, and GO/Fe_3_O_4_ after adsorption.

**Figure 5 nanomaterials-09-01429-f005:**
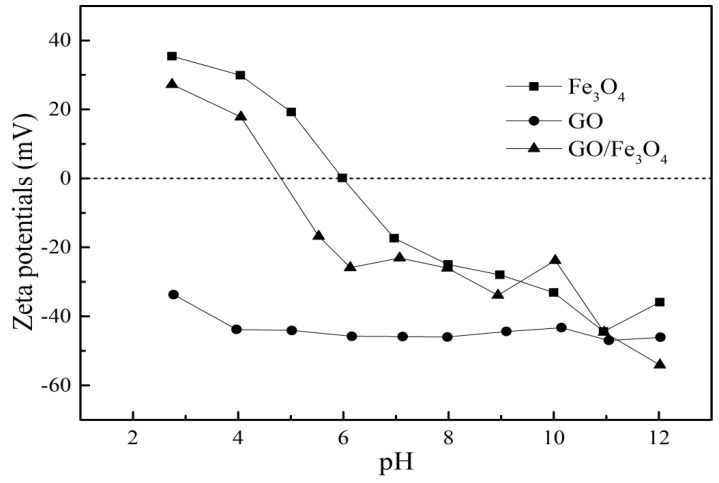
Zeta potential of GO, Fe_3_O_4_, and GO/Fe_3_O_4_.

**Figure 6 nanomaterials-09-01429-f006:**
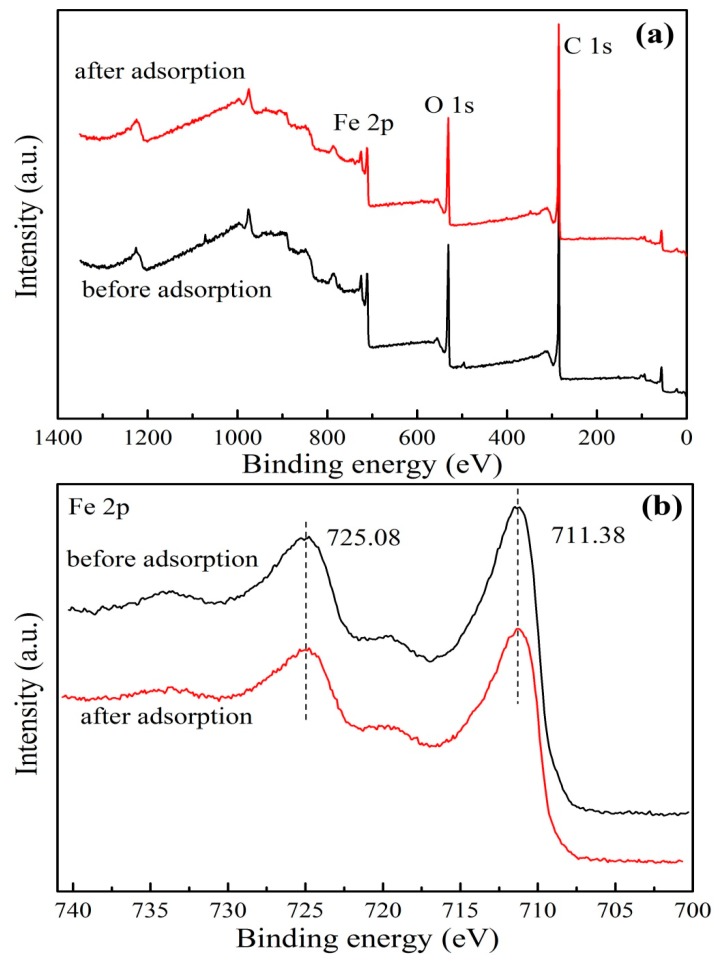
XPS spectra of: (**a**) the survey scan, and high-resolution scans of: (**b**) Fe2p, (**c**) C1s before adsorption, and (**d**) C 1s after adsorption.

**Figure 7 nanomaterials-09-01429-f007:**
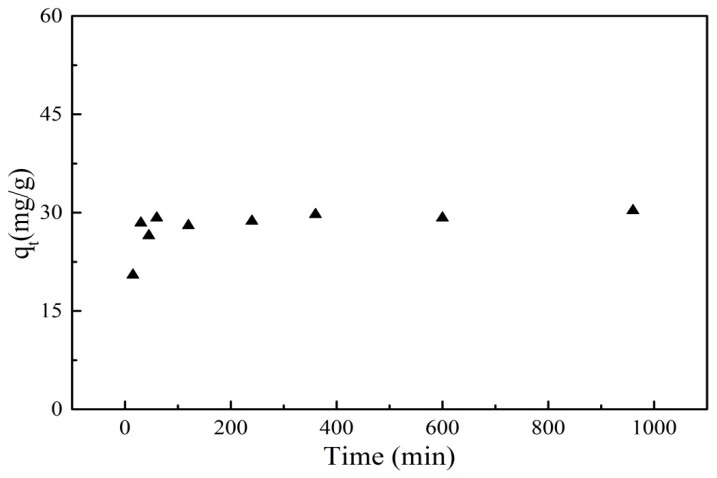
Effect of contact time on the adsorption of PRO onto GO/Fe_3_O_4_ (*C_0_* = 25 mg/L, *V* = 20 mL, *m* = 0.01 g, pH = 7.5).

**Figure 8 nanomaterials-09-01429-f008:**
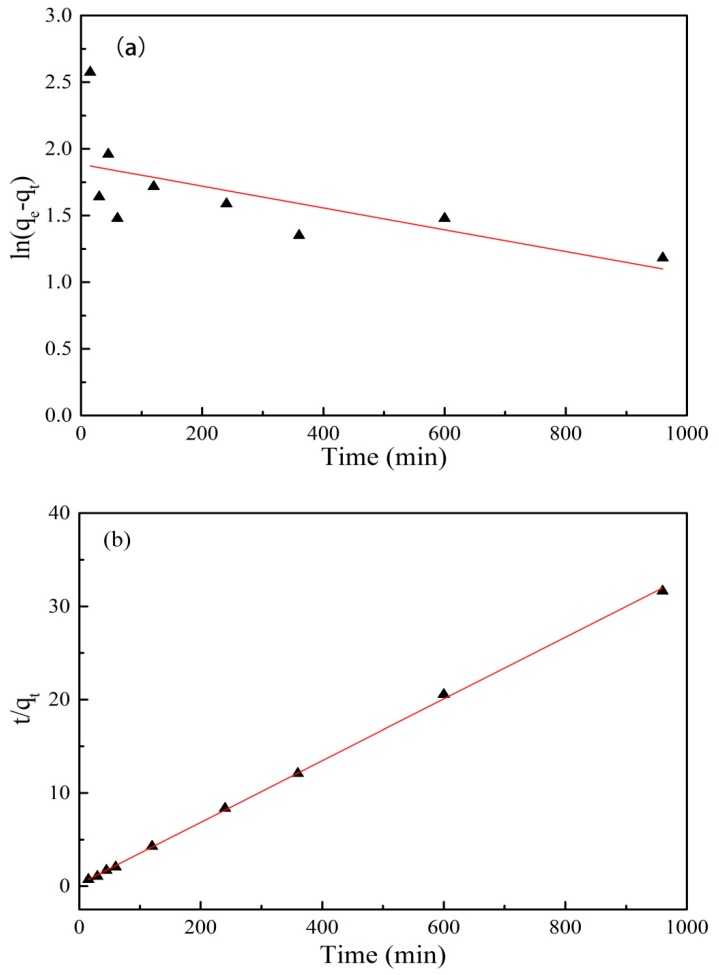
Adsorption kinetics curves of PRO onto GO/Fe_3_O_4_: (**a**) pseudo-first-order and (**b**) pseudo-second-order.

**Figure 9 nanomaterials-09-01429-f009:**
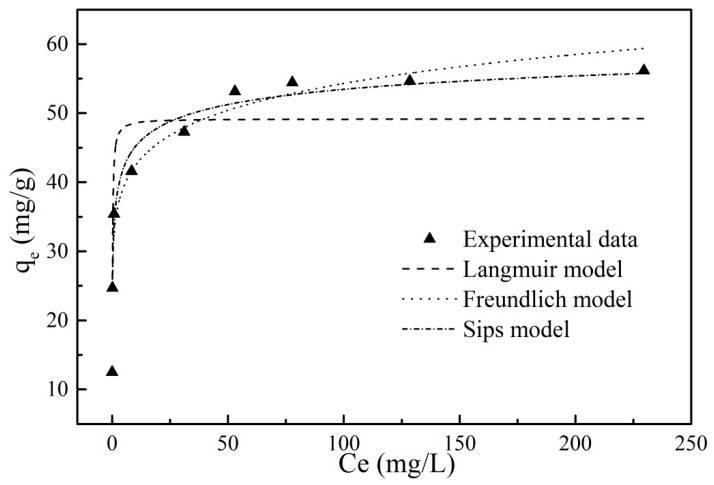
Langmuir, Freundlich, and Sips isotherm simulations for PRO onto GO/Fe_3_O_4_ (*t* = 24 h, *V* = 20 mL, *m* = 0.01 g, pH = 7.5).

**Figure 10 nanomaterials-09-01429-f010:**
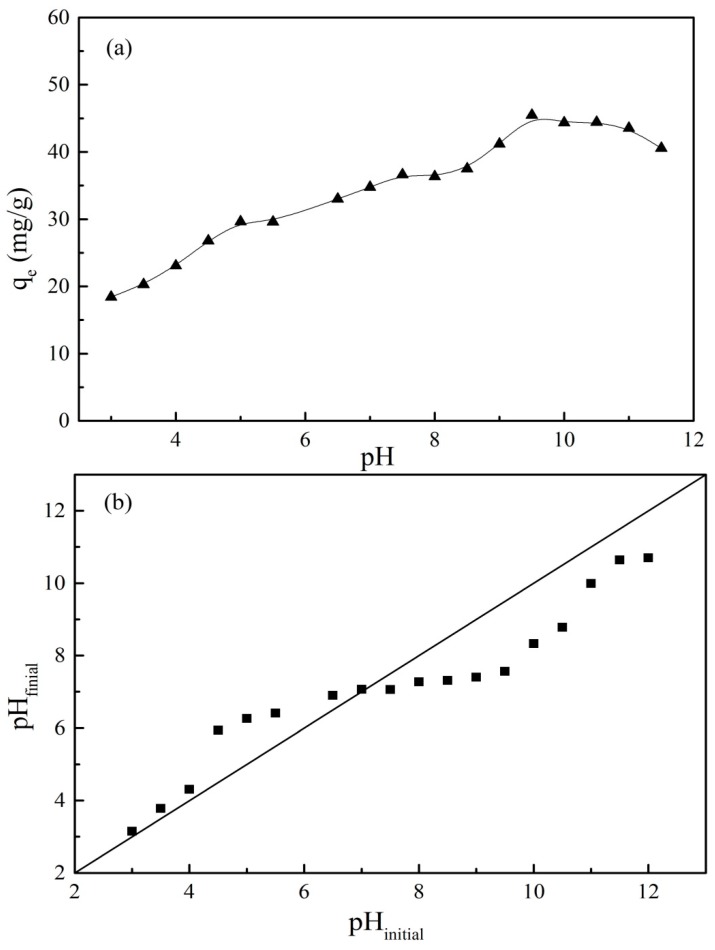
(**a**) Effect of pH on the adsorption of PRO onto GO/Fe_3_O_4_, and (**b**) the variation tendency of pH after the adsorption equilibrium was reached (*C_0_* = 25 mg/L, *V* = 20 mL, *m* = 0.01 g, *t* = 24 h).

**Figure 11 nanomaterials-09-01429-f011:**
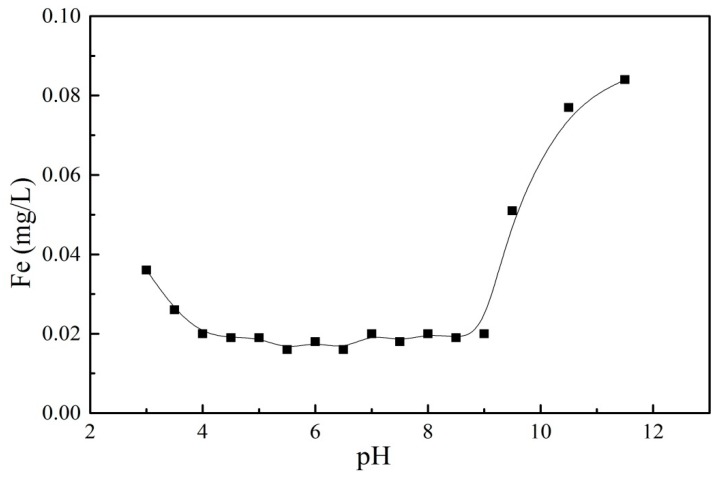
The leaching amount of Fe on GO/Fe_3_O_4_ at different pH levels.

**Figure 12 nanomaterials-09-01429-f012:**
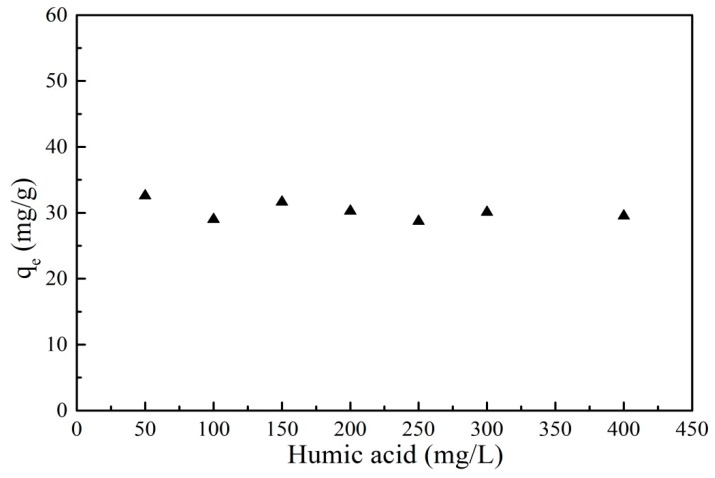
Effect of humic acid for PRO onto GO/Fe_3_O_4_ (*C_0_* = 25 mg/L, *V* = 20 mL, *m* = 0.01 g, *t* = 24 h, pH = 7.5).

**Table 1 nanomaterials-09-01429-t001:** Atomic percentages of C, O, Fe, and N.

Types	Atomic %			
	C 1s	O 1s	Fe 2p	N 1s
Before adsorption	66.54	22.04	7.7	0.47
After adsorption	71.19	19.32	5.19	0.87

**Table 2 nanomaterials-09-01429-t002:** Kinetics parameters for the adsorption of PRO onto GO/Fe_3_O_4_.

q_exp_ (mg/g)	Pseudo-First-Order Equation			Pseudo-Second-Order Equation		
	K_1_ (1/min)	q_cal_ (mg/g)	R^2^	K_2_ (g/(mg · min))	q_cal_ (mg/g)	R^2^
30.340	0.0008	6.580	0.338	0.0045	30.303	0.999

**Table 3 nanomaterials-09-01429-t003:** The parameters for Langmuir, Freundlich, and Sips isotherm models of PRO adsorption onto GO/Fe_3_O_4_.

Langmuir Equation			Freundlich Equation			Sips Equation			
q_m_ (mg/g)	K_L_ (L/mg)	R^2^	K_F_ (mg/g)	n	R^2^	q_m_ (mg/g)	b (L/mg)	n	R^2^
49.213	7.629	0.857	33.123	9.317	0.961	66.225	0.322	3.443	0.983

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
