# Peer review of "New Insights into the Interaction between Graphene Oxide and Beta-Blockers"

_nanomaterials, 2019, doi:10.3390/nano9101429_

Round 1
Reviewer 1 Report
The paper is well organized, the results are explained with details and they are coherently supported by the reported measurements. Therefore, the paper can be considered for the publication after the following minor revisions:
It is important to point out the original contribution of the authors. Please, compare these results with previous studies and report their new insights into this field. Add the error bars to the experimental data Add more detail about the magnetic property in terms of experimental tests, measurements, etc.
Author Response
Point: The paper is well organized, the results are explained with details and they are coherently supported by the reported measurements. Therefore, the paper can be considered for the publication after the following minor revisions. It is important to point out the original contribution of the authors. Please, compare these results with previous studies and report their new insights into this field. Add the error bars to the experimental data Add more detail about the magnetic property in terms of experimental tests, measurements, etc.
Response: Thank you very much for your approval of this paper. Thank you very much for your valuable suggestions. In the revised manuscript, we compared the adsorption properties of different adsorbents. The relevant content was as follows. To further evaluate the performance of the adsorbent, the adsorption capacity of GO/Fe3O4 was compared to other adsorbents. The adsorption capacity of PRO onto GO was 67 mg/g, and the adsorption capacity of GO/Fe3O4 was lower relatively due to the introduction of Fe3O4 nanoparticles. However, the loading of Fe3O4 greatly reduced the manufacturing cost of the adsorbent and facilitated the separation of GO from water. In the previous work, the adsorption of PRO onto acidified attapulgite (48.05 mg/g), chitosan modified attapulgite (26.38 mg/g), and coupling agent modified attapulgite (24.56 mg/g) were also studied. It was obvious that the adsorption capacity of PRO of GO/Fe3O4 was higher than the above adsorbents. In summary, GO/Fe3O4 can be used as a promising adsorbent to remove PRO from water. XPS characterization was added to the characterization section. The results showed that GO/Fe3O4 contained Fe3O4 nanoparticles before and after adsorption.
About error bars: When we were conducting experiments, all the data was repeated several times. If suspicious data appeared, retest. The final experimental data error was small and cannot be directly observed on the graph. Therefore, we took the average of experimental data to the graph without marking the error bar. (The isotherm added with the error bar was shown below)
Figure Isotherm adding error bars
Reviewer 2 Report
The paper from Deng et al reports on the application of the graphene chemistry in the absorbance of propanolol. The paper is well written and the results clearly reported. I think the paper should published as is._
Author Response
Point: The paper from Deng et al reports on the application of the graphene chemistry in the absorbance of propanolol. The paper is well written and the results clearly reported. I think the paper should published as is.
Response: Thank you very much for your approval of this paper.
Reviewer 3 Report
The paper proposed by Yuehua Deng et al. describes the preparation of magnetic graphene oxide nanocomposites for the adsorption of propranolol from water.
In general, the paper is well written and correctly discussed and the conclusions are well founded. Nevertheless, there are some lacks of novelty in the present form.
The authors claim the preparation of a new adsorbent for propranolol removal but the range of concentrations studied are out of the real values found in waste water. The Predicted Environmental Concentration (PEC) for propranolol is just 0.041 µg/L much lower than the range studied in this work (https://www.astrazeneca.com/content/dam/az/our-company/Sustainability/2017/Propranolol.pdf).
During the preparation of the new adsorbent several chemicals are used and their treatment for disposal is very expensive. Due to these facts, we can conclude that the proposed method is not environmentally friendly.
Any novelty concerning the functionalization of the GO obtained, is just a typical hydrothermal process.
Concerning the characterization of the adsorbent, there is a lack of techniques usually used, such as XPS, TPD, et.. in order to study the chemical nature of the surface of the adsorbent. Several techniques could be useful to confirm the explanation given for the adsorbent-adsorbate interactions. All the typical interactions: electrostatic attraction, hydrogen bonding and π-π interactions are compiled in the text. There is no discrimination of the different types.
There are no studies concerning the cyclability of the adsorbent. Due to the iron leaching is probable that the adsorbent loses its capacity after several cycles.
Finally, the most important point, there is no comparison between the results obtained in the present work with others papers previously published in order to know the effectiveness of the new material described.
Due to the previously exposed points I do not recommend the publication of the present paper in its actual form.
Author Response
Point 1: The authors claim the preparation of a new adsorbent for propranolol removal but the range of concentrations studied are out of the real values found in waste water. The Predicted Environmental Concentration (PEC) for propranolol is just 0.041 µg/L much lower than the range studied in this work.
Response 1: Thank you very much for your good comment. We notice that the concentration of propranolol in actual water bodies is lower than the concentration we studied. And in the next study we will pay attention to this point. The goal of the study was mainly on extending the application range of graphene oxide and investigating the mechanism of PRO uptake on GO/Fe3O4. Considering that graphene oxide has good dispersibility in water and is difficult to separate from water, we loaded Fe3O4 nanoparticles on its surface and achieving rapid separation by external magnetic field. And we are trying to provide a theoretical method for removing PRO by this adsorbent.
Point 2: During the preparation of the new adsorbent several chemicals are used and their treatment for disposal is very expensive. Due to these facts, we can conclude that the proposed method is not environmentally friendly.
Response 2: Thank you very much for your helpful suggestions. In subsequent experiments, we will continue to look for cheaper and more environmentally friendly methods of preparing materials. Our current work has begun to remove water pollutants by preparing straw biochar as an adsorbent. Reduce costs and achieve environmental friendliness through the recycling of crops.
Point 3: Any novelty concerning the functionalization of the GO obtained, is just a typical hydrothermal process.
Response 3: Thank you for your good comments. Considering the ease of preparation of magnetic graphene oxide, we have revised the title of the paper to "New insights into the interaction between graphene oxide and beta-bloker ".
Point 4: Concerning the characterization of the adsorbent, there is a lack of techniques usually used, such as XPS, TPD, et. in order to study the chemical nature of the surface of the adsorbent. Several techniques could be useful to confirm the explanation given for the adsorbent-adsorbate interactions. All the typical interactions: electrostatic attraction, hydrogen bonding and π-π interactions are compiled in the text. There is no discrimination of the different types.
Response 4: Thank you very much for your valuable suggestions. Based on our existing experimental conditions, we performed XPS characterization of the GO/Fe3O4 before and after adsorption. The specific content was stated in the last paragraph of Section 3.1. At the same time, Figure 7 XPS spectra of: (a) survey scan, and high-resolution scan of: (b) Fe2p; (c) C1s before adsorption, and (d) C 1s after adsorption and Table 1 The atomic percentage of C, O, Fe, and N were added in the paper. The text was as follows, The XPS spectrums of GO/Fe3O4 before and after adsorption of PRO with binding energy ranging from 0 to 1400 eV was acquired for identification of the surface elements and performance of a quantitative analysis. It was obvious in Figure 6 (a) that the peaks of Fe 2p, O 1s, and C 1s were shown in the full scan spectrums before and after adsorption of PRO, which suggested the existence of Fe, O, and C elements in the GO/Fe3O4. For the high-resolution Fe 2p of GO/Fe3O4 (Figure 6(b)), the binding energy of 711.38 and 725.38 eV were corresponded to Fe2+ and Fe3+, which indicated the successful synthesis of Fe3O4 nanoparticles and loading onto the surface of GO. The finding was consistent with above characterization results. From Figure 6 (c) and (d), the C=O, C-O and C=C/C-C characteristic bonds were presented at around 284.5, 285 and 286.5 eV, respectively. It can be observed that the peak intensity of C=C/C-C after adsorption became stronger than that of C=C/C-C before adsorption, which attributed to the introduction of PRO. The element content was exhibited in Table 1. It can be found that the carbon content after adsorption was significantly increased compared with before adsorption, which also indicated that PRO was successfully adsorbed onto GO/Fe3O4. The carbon element contained in PRO led to an increase in the total carbon content. The successful adsorption of PRO can also be illustrated by nitrogen element. The content of nitrogen increased from 0.47 to 0.87, which was caused by the amino group in PRO. At the same time, it can be seen that the content of Fe element was reduced, and it was speculated that the decrease of Fe element was caused by the leaching of Fe during the reaction, which was discussed in detail below. The increase in the content of other elements, especially the carbon element, led to a relative decrease in the proportion of oxygen elements.
Point 5: There are no studies concerning the cyclability of the adsorbent. Due to the iron leaching is probable that the adsorbent loses its capacity after several cycles.
Response 5: Thank you very much for your valuable suggestions. At the beginning of the experimental design, our goal was to expand the range of applications of graphene oxide. Although the adsorption capacity of graphene oxide is very high, the graphene oxide is not easily separated from the water after use, and may cause secondary pollution. With this in mind, we decided to load Fe3O4 nanoparticles on its surface, which was easy to solve the problem by external magnetic field.
After receiving your advice, we realized that the experimental design was not well thought out. Thus, cycling experiments were carried out to evaluate the availability of GO/Fe3O4. Methanol, deionized water, and 0.1 M HCl, were used to desorb PRO from GO/Fe3O4. The results of regeneration and reversibility investigation demonstrated that the desorption rate with methanol was higher than that with the other two solutions. Thus, methanol was chose for the eluent. However, after the first adsorption, the desorption rate with methanol desorption was only 33%. And after three cycles of experiments, the measured adsorption rate only reached 64% of the first adsorption rate. There may be some reasons for such results. Due to limited time, a suitable elution method had not been found and the desorption rate cannot be improved, resulting in a worse performance of adsorbent regeneration. On the other hand, it may be that the prepared adsorbent had a very strong binding ability to PRO, which caused PRO to not be easily desorbed from the adsorbent.
Thank you very much for your helpful suggestions, which is very helpful for our follow-up experiments. In subsequent experiments we will continue to find suitable desorption methods to improve the recyclability of the adsorbent and develop materials that are easier to recycle.
Point 6: Finally, the most important point, there is no comparison between the results obtained in the present work with others papers previously published in order to know the effectiveness of the new material described.
Response 6: Thank you very much for your helpful suggestions. After listening to your suggestions, we compared the adsorption capacities of different adsorbents. The details are as follows. To further evaluate the performance of the adsorbent, the adsorption capacity of GO/Fe3O4 was compared to other adsorbents. The adsorption capacity of PRO onto GO was 67 mg/g, and the adsorption capacity of GO/Fe3O4 was lower relatively due to the introduction of Fe3O4 nanoparticles. However, the loading of Fe3O4 greatly reduced the manufacturing cost of the adsorbent and facilitated the separation of GO from water. In the previous work, the adsorption of PRO onto acidified attapulgite (48.05 mg/g), chitosan modified attapulgite (26.38 mg/g), and coupling agent modified attapulgite (24.56 mg/g) were also studied. It was obvious that the adsorption capacity of PRO of GO/Fe3O4 was higher than the above adsorbents. In summary, GO/Fe3O4 can be used as a promising adsorbent to remove PRO from water.

Round 2
Reviewer 3 Report
Authors has addressed correctly, practically, all the points given in my first evaluation. Nevertheless, there is a point which must be discussed more extensively: that is the comparison of the results obtained in this paper and those previously published.
This last point regarding the reference 27 in the new version is the most important one. After a careful reading of the aforementioned reference, the reviewer had a doubt related to the originality of this paper. There are many similarities between both papers and, before publication, the main differences should be pointed out.
Author Response
Response to Reviewer Comments
Point: Authors has addressed correctly, practically, all the points given in my first evaluation. Nevertheless, there is a point which must be discussed more extensively: that is the comparison of the results obtained in this paper and those previously published.
This last point regarding the reference 27 in the new version is the most important one. After a careful reading of the aforementioned reference, the reviewer had a doubt related to the originality of this paper. There are many similarities between both papers and, before publication, the main differences should be pointed out.
Response: Thank you very much for your meaningful evaluation. Thank you very much for reading our previously published paper, "Fast Removal of Propranolol from Water by Attapulgite/Graphene Oxide Magnetic Ternary Composites".
In our previous study, attapulgite (ATP) had been studied as an adsorbent to remove propranolol (PRO) from water. It was found that ATP had a certain removal effect on PRO, but its settling speed was so fast that it cannot be in full contact with contaminants. Meanwhile, ATP was not easy to separate from water after adsorbing pollutants. Taking these two points into consideration, Fe3O4 particles were loaded on the ATP surface to solve the problem of separation difficulties by applying an external magnetic field. Furthermore, grapheme oxide (GO) was more dispersible in water, so GO was used as a dispersing agent to provide resistance for ATP settling. Kyzas et al. indicated that GO had a better adsorption effect on PRO. Therefore, the addition of GO as an auxiliary material can both disperse ATP and adsorb PRO as an adsorbent.
During the experiment, it was found that the adsorption adsorption of the material added with GO was greatly improved. Combined with the adsorption of PRO on GO, we considered whether it would be better to combine GO with Fe3O4 particles without ATP, which not only overcame the disadvantage that GO was difficult to separate from water, but also maintained a relatively high adsorption capacity. Experimental results showed that the adsorption capacity of magnetic graphene oxide (GO/Fe3O4) was indeed higher than that of ternary composite. At the same time, the influencing factors of the binary composite preparation process are reduced compared to the ternary composites. Further, we found that the experimental conditions were not very sufficient when Kyzas et al. conducted the adsorption experiment of GO on PRO. On this basis, we increased the research conditions and better analyzed the adsorption mechanism of PRO on magnetic GO, which expanded the scope of application of GO.
